# The Mechanism of Action of SAAP-148 Antimicrobial Peptide as Studied with NMR and Molecular Dynamics Simulations

**DOI:** 10.3390/pharmaceutics15030761

**Published:** 2023-02-24

**Authors:** Morgane Adélaïde, Evgeniy Salnikov, Francisco Ramos-Martín, Christopher Aisenbrey, Catherine Sarazin, Burkhard Bechinger, Nicola D’Amelio

**Affiliations:** 1Unité de Génie Enzymatique et Cellulaire UMR 7025 CNRS, Université de Picardie Jules Verne, 80039 Amiens, France; 2Institut de Chimie, UMR7177, Université de Strasbourg/CNRS, 67000 Strasbourg, France

**Keywords:** antimicrobial, peptides, biomembranes, NMR, molecular dynamics, paramagnetic relaxation enhancement, solid-state NMR spectroscopy, membrane topology

## Abstract

Background: SAAP-148 is an antimicrobial peptide derived from LL-37. It exhibits excellent activity against drug-resistant bacteria and biofilms while resisting degradation in physiological conditions. Despite its optimal pharmacological properties, its mechanism of action at the molecular level has not been explored. Methods: The structural properties of SAAP-148 and its interaction with phospholipid membranes mimicking mammalian and bacterial cells were studied using liquid and solid-state NMR spectroscopy as well as molecular dynamics simulations. Results: SAAP-148 is partially structured in solution and stabilizes its helical conformation when interacting with DPC micelles. The orientation of the helix within the micelles was defined by paramagnetic relaxation enhancements and found similar to that obtained using solid-state NMR, where the tilt and pitch angles were determined based on ^15^N chemical shift in oriented models of bacterial membranes (POPE/POPG). Molecular dynamic simulations revealed that SAAP-148 approaches the bacterial membrane by forming salt bridges between lysine and arginine residues and lipid phosphate groups while interacting minimally with mammalian models containing POPC and cholesterol. Conclusions: SAAP-148 stabilizes its helical fold onto bacterial-like membranes, placing its helix axis almost perpendicular to the surface normal, thus probably acting by a carpet-like mechanism on the bacterial membrane rather than forming well-defined pores.

## 1. Introduction

The development of new antimicrobial agents is one of the main challenges of our times in view of the increasing threat of bacterial resistance. Multidrug-resistant (MDR) organisms expose a constantly increasing number of patients to untreatable infections [1,2,3]. Recalcitrant bacterial strains are often caused by nosocomial infections, many of them involving the so-called ESKAPE bacteria: *Enterococcus faecium*, *Staphylococcus aureus*, *Klebsiella pneumoniae*, *Acinetobacter baumannii, Pseudomonas aeruginosa*, and *Enterobacter* spp., some of which are able to form dangerous biofilms [3,4,5,6].

Antimicrobial peptides (AMPs) are promising molecules in the quest for new antibiotics. Produced by several organisms in nature, AMPs are relatively short (10–30 residues) and exhibit a wide range of antimicrobial or immunomodulating activities [7,8,9]. In contrast to standard antibiotics, many AMPs are able to quickly permeate cell membranes causing irreversible damage [7,10,11]. For this reason, they can be used in synergy with antibiotics to grant intracellular access to the latter, thus overcoming resistance based on reduced permeabilization of the bacterial membrane [12,13]. Some AMPs are able to bypass the bacterial membrane without damaging it and exert their antimicrobial action intracellularly [7,10,11,14,15,16,17,18]. While exceptions exist [19], AMPs are less prone to resistance due to multiple factors. First, their fast-killing rate (compared to antibiotics) does not favor bacterial replication, possibly leading to new resistance [7,20,21]. Second, the heterogeneity of their target (bacterial membrane) makes the interaction less specific, thus making it harder for bacteria to completely impair their action by single-point mutations [22]. Third, some AMPs act by multiple modes of action [23,24,25].

One of the main limitations of AMPs is their degradation or loss of activity in blood and tissues [3,26]. Several strategies have been used to overcome these problems such as the substitution of L-amino acids with their D-isomers or the introduction of non-natural amino acids [27,28,29,30]. Other strategies are based on the study of a huge portfolio of peptides designed from an existing one, aiming to optimize its activity and reduce its degradation properties. One of these peptides is SAAP-148 (Synthetic Antimicrobial and Antibiofilm Peptide), which derives from LL-37 and shows an excellent activity profile against drug-resistant bacteria and biofilms while resisting degradation when tested in physiological conditions [31,32,33]. Together with SAAP-148, another 25 peptides were inspired by LL-37, all designed using the substitution of anionic glutamine with cationic arginines or lysines, thus enhancing their cationicity and helicity in order to increase their antimicrobial activities. When tested against *Staphylococcus aureus*, SAAP-148 performed better than others with an LC_99.9_ of 1.6 μM in PBS and 12.8 μΜ in the presence of plasma in the medium [31]. Similar values were obtained using other resistant strains such as *Pseudomonas aeruginosa*, *Enterococcus faecium*, *Klebsiella pneumoniae*, *Acinetobacter baumannii* [31], or *Enterococcus hirae* [33]. Despite its optimal pharmacological properties (without remarkable toxic effects) [31,32,33], its mechanism of action remains poorly understood at the molecular level. The aim of this work is to elucidate its interaction with bacterial and mammalian membranes at the molecular level of detail.

The interaction of AMPs with membranes has been described in detail using the SMART model [34], which shows how amphipathic peptides tend to align to the surface and intercalate among the lipid headgroups, while apolar peptides tend to insert and span the thickness of the bilayer. The model suggests that bilayers adapt to membrane-inserted peptides until threshold concentrations are reached. In the presence of small headgroups, such as ethanolamine of PE, the amphipathic peptide may even stabilize the bilayer properties by intercalation. However, at high peptide-to-lipid ratios, the peptide tends to cover the surface (carpet model) causing the formation of transient micro-sized pores [35,36,37,38,39,40,41,42,43,44,45] that allow the transmembrane passage of molecules [11,46,47]. This is possible because of the “soft” nature of the bilayer which can deform, change its thickness, and adapt to different perturbations [48,49]. Depending on the nature of the peptide and the conditions (pH, buffer, temperature, etc.), different structures can be formed including barrel-stave or toroidal (also called “worm-hole”) pores, induction of non-lamellar phases, non-lytic depolarization, localized thinning bicelle formation, and detergent-type micellization which efficiently dissolve the membrane [7,11,35,48].

## 2. Materials and Methods

### 2.1. Peptide Synthesis

The peptide SAAP-148 (Ac-LKRVWKRVFKLLKRYWRQLKKPVR-NH_2_) was synthesized following the standard Fmoc-synthesis protocol using a Millipore 9050 automatic peptide synthesizer. The carboxyl amide C-terminus was archived using TentaGel S Ram resin (Rapp Polymeres, Tübingen Germany). Acetylation of the N-terminus was performed using 2% acetic anhydride. The leucine at position 11 or at position 12, one at a time, were ^15^N labeled using labeled Fmoc Leucine (Cortecnet, Voisins le Bretonneux, France). The peptides were purified with reverse phase HPLC (Gilson, Villiers-le-bel, France) using a preparative C18 column (Luna, C-18-100 Å-5 mm, Phenomenex, Le Pecq, France) and an acetonitrile/water gradient (Appendix A). Their identity and purity (>90%) were checked using MALDI mass spectrometry (MALDI-TOF Autoflex, Bruker Daltonics, Bremen, Germany) (Appendix A).

### 2.2. Liquid-State NMR Spectroscopy and Sample Preparation

#### 2.2.1. Titration of SAAP-148 with DPC Micelles

SAAP-148 was solubilized at a concentration of 2.4 mM in 500 µL of 50 mM phosphate buffer at pH 6.6 containing 10% of D_2_O and 0.1 mM deuterated sodium 3-(trimethylsilyl)propionate:d4 (TSP:d_4_) as an internal reference. A few microliters of a 1 M solution of dodecylphosphocholine-d38 (DPC:d38) were gradually added to obtain a final concentration of 60 mM at 310 K.

#### 2.2.2. Paramagnetic NMR Experiment

A measured amount of 16-doxyl stearic acid (16-DSA) was dissolved in deuterated methanol and added to samples of SAAP-148 in DPC micelles in microliter amounts until the paramagnetic effect was observed as a wider linewidth on selected peaks (typically in the ratio 16-DSA:DPC 1:10). Paramagnetic contributions were evaluated as percentage loss in the intensity of Hα/Cα cross peaks in ^1^H,^13^C-HMQC spectra. Errors were evaluated based on spectral noise using the relation for the propagation in ratios A = $\frac{B}{C}$: ΔA = A($\frac{\Delta B}{B}+\frac{\Delta C}{C}$), where B is the cross-peak intensity without 16-DSA, C is the cross-peak intensity with 16-DSA, and ΔB and ΔC are the spectral noise. Experiments were run at 310 K.

#### 2.2.3. Titration of SAAP-148 with Bicelles

SAAP-148 was solubilized in 500 µL of 50 mM phosphate buffer at pH 6.6 containing 10% of D_2_O and 0.1 mM TSP-d_4_ as an internal reference. Samples were typically at a concentration of 0.8 mM. A 1 M solution of isotropic bicelles was obtained by solubilizing in chloroform 1,2-dihexanoyl-sn-glycero-3-phosphocholine (DHPC), 1,2-dimyristoyl-sn-glycero-3- phosphocholine (DMPC), and 1,2-dimyristoyl-sn-glycero-3-phospho-(1′-rac-glycerol) (DMPG) at molar ratios 66.7:25.0:8.3. Few microliters of a 1 M solution of DHPC/DMPC/DMPG bicelles were gradually added to obtain a final concentration of 50 mM at 310 K.

#### 2.2.4. NMR Acquisition and Processing

Liquid state NMR experiments were recorded using a 500 MHz Bruker spectrometer equipped with a 5 mm Broadband Inverse (BBI) probe. Backbone resonance assignments were achieved by ^1^H,^13^C-HMQC, ^1^H,^1^H-TOCSY (mixing time of 60 ms), and ^1^H,^1^H-NOESY (mixing time of 200 ms). Spectra were analyzed with TopSpin 4 (Bruker BioSpin). Secondary structure prediction was achieved by observing deviations from random coil values obtained with the POTENCI web server accessed on 21 October 2021 (https://st-protein02.chem.au.dk/potenci/) [50] at 310 K.

### 2.3. Solid-State NMR Spectroscopy and Sample Preparation

The sample preparation and NMR measurements here are described in detail and illustrated in [51]. In short, oriented samples for solid-state NMR were prepared by co-dissolving 3.25 mg of SAAP-148 powder in methanol/chloroform (1:1 by volume) with corresponding amounts of POPE (1-palmitoyl-2-oleoyl-sn-glycero-3-phosphoethanolamine) and POPG (1-palmitoyl-2-oleoyl-sn-glycero-3-phospho-(1′-rac-glycerol)) lipids to reach a 2:75:25 molar ratio. The sample solution was vortexed and sonicated using a bath sonicator for 5 min and the solvent was partially evaporated under a stream of nitrogen gas. The viscous mixture was spread on 22 ultrathin cover glass plates (8 × 12 mm^2^, Marienfeld, Lauda-Königshofen, Germany), and the samples were dried under vacuum overnight to remove the organic solvent. Thereafter, the glass plates were incubated at 96% relative humidity at 37 °C (i.e., well above the gel-to-fluid transition temperature) for two days, stacked on top of each other, tightly sealed and introduced in the NMR spectrometer with the glass plates normal being parallel to external magnetic field direction.

The preparation of non-oriented samples started by mixing POPE/POPG-d_31_ (3:1, 4 mg/1.5 mg) or POPE/POPE-d_31_/POPG (2:1:1, 2.9 mg/1.5 mg/1.5 mg) and the appropriate amount of peptide to reach a 2% mole ratio in methanol/chloroform 1/1 (*v*/*v*). The sample solution was vortexed and sonicated using a bath sonicator for 5 min and the solvent was evaporated using exposure to a stream of nitrogen and to a high vacuum overnight in such a manner to form a film along the walls of a glass tube. The sample was resuspended in 23.5 μL (water content h = 0.81) of 100 mM Tris buffer (pH 7.4) and involved vortexing and bath sonication, as well as 3 freeze/heat cycles at −20 °C and 40 °C. The glass tube (6 mm outer diameter) with the sample was inserted into the solenoidal coil of the solid-state NMR probe.

Proton-decoupled ^31^P solid-state NMR spectra were acquired at 121.575 MHz using a Bruker Avance wide-bore 300 solid-state NMR spectrometer equipped with a commercial double-resonance flat-coil probe (Bruker, Rheinstetten, Germany) [52]. A Hahn-echo pulse sequence [53] was used with a π/2 pulse of 5 µs, a spectral width of 100 kHz, an echo delay of 100 µs, an acquisition time of 10.2 ms, and a recycle delay of 3 s. External 85% H_3_PO_4_ at 0 ppm was used for calibration. The temperature was set to 310 K.

The proton-decoupled ^15^N cross-polarization spectra of static aligned samples were acquired at 30.43 MHz using a Bruker Avance wide bore 300 NMR spectrometer. An adiabatic CP pulse sequence was used with a spectral width, acquisition time, CP contact time, and recycle delay times of 25 kHz, 10.2 ms, 0.4 ms, and 3 s, respectively [51,54]. The ^1^H π/2 pulse and spinal-64 heteronuclear decoupling field strengths B_1_ corresponded to a nutation frequency of 31 kHz. Then, 40 k scans were accumulated, and the spectra were zero-filled to 4 k points. An exponential line broadening of 50 Hz was applied before the Fourier transformation. Spectra were externally referenced to ^15^NH_4_Cl at 39.3 ppm [55]. Samples were cooled with a stream of air at a temperature of 310 K.

Next, ^2^H solid-state NMR spectra of deuterated lipid samples were acquired using a quadrupole pulse-echo sequence [56] with a recycle delay of 0.3 s, an echo time of 100 μs, a dwell time of 0.5 μs, and a π/2 pulse of 5 µs. Before the Fourier transformation of the free induction decay, an exponential apodization with a line broadening of 100 Hz was applied. Spectra were externally referenced to ^2^H_2_O (0 Hz). The temperature was set to 310 K.

The deuterium order parameters (S_CD_) for each CD_2_ and CD_3_ group were determined according to $S_{CD}^{i}=\frac{4}{3}\;\frac{h}{e^{2}qQ}\Delta^{i}\upsilon$, where Δ^i^υ is the quadrupolar splitting of segment i and (e^2^qQ/h) is the static quadrupole coupling constant (167 kHz) observed for deuterons within C–D bonds [57].

Orientational restraints from oriented solid-state NMR spectra. A coordinate system was defined where the tilt angle is defined as the angle between the helix long axis and the membrane normal. The α-helical conformation had Ramachandran angles of (ϕ = −65°, ψ = −45°). The tilt and pitch angles were scanned, where at each orientation, the corresponding ^15^N chemical shift was calculated. The standard deviation (SD) of a Gaussian line shape takes into account orientational heterogeneity. Independent wobbling (10° Gaussian distribution) and azimuthal fluctuations around the helix long axis (SD 18°) were taken into consideration by averaging the resonance values on the ensemble of orientations with the corresponding Gaussian distributions, similar to the dynamics of surface-bound sequences that have been tested previously including N-terminal 17 residues domain of Huntingtin protein [58].

### 2.4. Circular Dichroism

The peptides were dissolved at 50 µM in a 10 mM phosphate buffer at pH 7 in the presence and in the absence of NaCl 100 mM or Tris buffer at pH 7. CD spectra were recorded at 37 °C using a J-810 spectropolarimeter (Jasco, Tokyo, Japan) using 2 scans, a step resolution of 1 nm, and an adaptive integration time between 1 and 8 s. The path length of the quartz cuvette was 1 mm. The spectra were recorded between λ = 190 and 250 nm.

### 2.5. Molecular Dynamics Simulations

The CHARMM-GUI web server [59,60,61] was used to prepare the bilayer systems for MD simulations. GROMACS software [62] was used for the calculations with the CHARMM36m force field [63]. Water molecules were described with the TIP3P model [64]. Bilayers contained 64 lipids in each leaflet, and peptide molecules (whose initial structure was calculated using I-Tasser [65]) were placed over the upper leaflet at a distance >10 Å, avoiding intermolecular contacts in the setups containing more than one molecule of the peptide. The C-terminus was amidated with the CHARMM terminal group functionality. Lysine and arginine side chains were protonated. A 50 Å thick water layer was added above and below the membrane resulting in about 15,000 water molecules (30,000 in the case of pure cardiolipin (CL) systems). A concentration of 150 mM of NaCl, CaCl_2_, or MgCl_2_ was used when appropriate.

Periodic boundary conditions were applied for all the simulations. The particle mesh Ewald (PME) method was used for long-range electrostatic interactions. Each system was energy-minimized with a 5000-step steepest-descent algorithm. Equilibration was performed with the Berendsen barostat [66] while a Parrinello–Rahman barostat [67] was used to maintain pressure (1 bar) semi-isotropically with a time constant of 5 ps and a compressibility of 4.5 × 10^–5^ bar^–1^. A Nose–Hoover thermostat [68,69] with a time constant of 1 ps was chosen to maintain the systems at 310.15 K. All bonds were constrained using the LINear Constraint Solver (LINCS) algorithm, allowing a 2 fs integration step. Then, 500 ns simulations were run (except when a different duration is specified in the text) after the standard CHARMM-GUI minimization and equilibration steps [70]. The whole process (minimization, equilibration, and production run) was performed twice in the absence of peptides and thrice in their presence. Convergence was assessed using the analysis of RMSD and polar contacts.

The following molar proportions were used for the mixed bilayers: POPE/POPG 70/30, POPE/POPG/TOCL2 67/27/6, and POPC/CHOL 70/30. MD trajectories were analyzed using GROMACS tools. Graphs and images were created with GNUplot [71] and PyMol [72]. MOLMOL [73] and VMD [74] were used for visualization.

## 3. Results and Discussion

### 3.1. Liquid-State NMR studies of SAAP-148 Interacting with Model Membranes

#### 3.1.1. SAAP-148 in Aqueous Solution

The ^1^H NMR spectrum of SAAP-148 in phosphate buffer at pH 6.6 is shown in Figure 1A (bottom spectrum). The significant broadening of signals might be due to exchange with the water solvent (for exchangeable protons) and/or conformational exchange/aggregation equilibria that occur on times scales equivalent to the acquisition time (micro- to milliseconds). The latter hypothesis is supported by the fact that broadening is observed also in the aromatic and aliphatic region that does not contain exchangeable protons. The concentration dependence of the ^1^H NMR spectrum (Appendix A) strongly suggests that SAAP-148 is aggregating or rather forming a multimer at concentrations larger than about 100 µM. In helical conformation, this peptide would form a perfectly amphipathic helix, suggesting that it might dimerize to avoid exposure of the apolar side chain to the water. At concentrations lower than 100 µM, the NMR lines are much narrower and CD spectra (Appendix A) show the characteristic pattern of a random coil (minimum at about 200 nm).

Even though extreme broadening in the amide region prevents resonance assignment by ^1^H,^1^H-TOCSY and ^1^H,^1^H-NOESY at the concentration used for NMR, the signal dispersion appears compatible with residual structuring. A value of 0.88 ppm is in fact intermediate between a predicted dispersion of 0.48 ppm for a random coil (estimated usingPOTENCI web server [50], Appendix A) and that of a full alpha-helical structure (1.05 ppm as estimated using SHIFTX2 web server [75], Appendix A). However, the absence of HN-HN cross peaks in the NOESY spectrum indicates that such structures, if helical, are not well defined in solution.

#### 3.1.2. SAAP-148 Interaction with Biomimetic Membrane Models

The addition of a concentrated solution of DPC-micelles significantly modifies the spectrum of SAAP-148. Many peaks re-emerge from the baseline at a protein:detergent molar ratio of 1:25 (see, for example, the two HN protons of the tryptophan side chain at about 10 ppm in Figure 1A), and the final spectrum in micelles become relatively well resolved to allow resonance assignment (Figure 1B). The gradual shift and broadening of SAAP-148 peaks as DPC micelles are added to the solution are indicative of fast exchange close to coalescence on the NMR time scale.

The main limitation of DPC resides in the fact that micelles do not form bilayers and that access to the lipid core is facilitated by their high curvature [76]. A better membrane model is provided with bicelles constituted by a mixture of DHPC and DMPC where the bilayer formed by DMPC is stabilized by the short chains of DHPC, which prevent direct exposure of the DMPC acyl chains to water [77]. Furthermore, these systems can incorporate to some extent different headgroups for modeling various types of biological membranes (e.g., PG for bacteria [78,79]). For solution-state NMR, isotropic bicelles are ideal for their fast-tumbling properties [77]. For this reason, we titrated a solution of SAAP-148 with a concentrated solution of isotropic bicelles constituted by a mixture of DHPC/DMPC/DMPG at a ratio of 2:0.25:0.75 (Figure 1C). The comparison of Figure 1A,C suggests that SAAP-148 interacts in a similar fashion with micelles and isotropic bicelles. Unfortunately, the size of the complex results in a shortening of the transversal relaxation time, which prevents the observation of H_α_/C_α_ resonances in the ^1^H,^13^C-HSQC (or ^1^H,^13^C-HMQC) spectra and consequently the assignment of the bound form of SAAP-148. Interestingly, Figure 1C shows that the binding process takes place with a slower exchange than with DPC micelles (peaks do not shift gradually but tend to reappear at different positions along the titration), indicating a stronger interaction.

The full ^1^H backbone assignment was achieved for SAAP-148 in DPC micelles using the 2D ^1^H,^1^H-TOCSY, ^1^H,^1^H-NOESY, and ^1^H,^13^C-HMQC spectra (Figure 1B and Appendix A), allowing the calculation of the Hα and Cα resonance deviations from random coil values (Figure 2A,B). Persistent negative (Hα) and positive (Cα) deviations beyond the threshold limit (0.1 ppm for ^1^H and 0.7 ppm for ^13^C) [80] indicate the formation of a stable alpha helix interrupted only by proline 22 at the C-terminus [81,82,83].

By introducing a paramagnetic species in the system, the formation of a stable alpha-helix and its orientation in the micelles can be determined. Paramagnetic probes are commonly used in NMR measurements for their strong peak-broadening effect, which depends on the sixth power of the distance [84,85,86]. They can serve as a reference point to determine the location of atoms in the species under study. In our case, we added 16-doxyl stearate (16-DSA) at a molar ratio of 16-DSA:DPC 1:10. This is a stearic acid bearing a paramagnetic spin label at the end of its acyl chain. This location allows placing the paramagnetic probe in the core of the micelles once the molecule gets inserted [84]. Paramagnetic effects were quantified by the percentage loss in the intensity of Hα/Cα cross peaks in the ^1^H,^13^C-HMQC spectra. Figure 2C shows that the paramagnetic effect approximately follows the periodicity of a helix (3.6 amino acid per turn), thus confirming the results obtained with the analysis of chemical shift deviations (Figure 2A,B). Interestingly, the maxima in Figure 2C have comparable intensity along the sequence, an effect compatible with an orientation of the helix axis parallel to the membrane surface (or to the tangent touching it, given the high curvature of the micelle). A more pronounced paramagnetic effect at the N-terminus suggests its slightly deeper insertion in the micelle. Furthermore, positively charged Lys and Arg residues are clearly less affected than the others, indicating that the amphipathic helix (see insert in Figure 2C) exposes its apolar face towards the center of the micelle. Assuming that lysine side chains interact with phosphate groups of phospholipids (as often observed for antimicrobial peptides [87]), this implies that the peptide is well inserted in the micelle with its apolar face making important van der Waals contact with lipid chains.

### 3.2. Solid-State NMR Studies on SAAP-148 Peptide Interacting with Model Membranes

#### 3.2.1. The Orientation of SAAP-148 in Bacterial Biomimetic Membranes

While NMR data in solution have provided important information on the interaction of SAAP-148 with DPC micelles, a better model of bacterial membranes can be obtained from phospholipid bilayers, which can be arranged either as liposomes or as planar structures on mechanical supports [51]. The large size of such systems requires the use of solid-state NMR spectroscopy in combination with isotopic labeling of either the peptide and/or the lipids. In order to confirm the results obtained with solution NMR spectroscopy also for more realistic systems, we have studied the interaction of SAAP-148 selectively labeled with ^15^N with POPE/POPG uniaxially oriented bilayers. Our choice is justified by the fact that bacterial cell membranes usually expose anionic lipids such as PG and CL and, in most gram-negative species, PE is the major phospholipid exposed to the outer leaflet of the cytoplasmic membrane [78,79].

The anisotropy of the ^15^N chemical shift is used to determine the pitch and tilt angles of the helix formed by SAAP-148 in oriented systems. These two parameters refer to the angles formed by the long (tilt) and short (pitch) cylindrical helix axes with the bilayer normal (which is assumed to be along the magnetic field direction in oriented systems). The homogeneity of the lipid orientation, parallel to the magnetic field, was ascertained using ^31^P solid-state NMR spectra (Figure 3A,B). The predominant intensity around 30 ppm is indicative that most lipids are well aligned along the surfaces of the mechanical support while some intensity extends up to −15 ppm, suggesting a global deformation of the lipid bilayer concomitant with other conformations and/or alignments of the lipid head group [89].

In a next step, peptides labeled with ^15^N in the central part of the peptide were investigated. This region adopts a stable helical conformation in the presence of detergent micelles (Figure 2) suggesting that Leu 11 and 12 are also part of this structured domain when interacting with lipid membranes. Indeed, when reconstituted into uniaxially oriented phospholipid bilayers the ^15^N spectra show well defined resonances indicative that both residues are part of a helical structure that is homogeneously aligned relative to the membrane normal (Figure 3C,D). The ^15^N chemical shifts are 80.0 ± 2.5 ppm and 70.0 ± 4.0 ppm, respectively, indicating a helix alignment approximately parallel to the membrane surface [90].

When analyzed in quantitative detail, these values are compatible with the combinations of tilt and pitch angles shown in Figure 3E as blue (Leu 11) and green lines (Leu 12), respectively. However, only two possible solutions (I and II) are compatible with both measurements and are highlighted in red in panel E of Figure 3. The two regions of tilt/pitch angular pairs correspond to the orientations represented in panel F. Despite the different conditions (DPC micelles vs. POPE/POPG bilayers), solution II is compatible with what was observed in the liquid state. Additionally, in that case, a slight inclination of the N-terminus towards the membrane core was observed (tilt), together with the orientation of apolar residues towards the interior of the bilayer (pitch).

#### 3.2.2. The Effect of SAAP-148 on Biomimetic Membrane Dynamics

Once the orientation of the peptide was determined in our membrane models, the effect of SAAP-148 on the phospholipids and their order parameters was also investigated. This was achieved using ^2^H static solid-state NMR spectroscopy of liposomes where either POPG or POPE were ^2^H labeled along the palmitoyl chain (Figure 4A,B). The resulting ^2^H solid-state NMR spectra are characterized by superimposed quadrupolar splittings, which represent the order parameter of the various C–D bond vectors [91,92]. This implies that it is possible to calculate the order parameter of each position of the deuterated chain by measuring the corresponding quadrupolar splitting in the spectrum [91,92] (Figure 4C,D).

The presence of SAAP-148 clearly affects the spectrum of POPE/POPG liposomes, independently of the ^2^H labeled species (POPG in panels A, C, and E or POPE in panels B, D, and F), thus ruling out the specificity for potential inhomogeneous phospholipid domains in the membrane. When focusing on the effect on the order parameter (Figure 4B–E), SAAP-148 perturbs all the lipid palmitoyl chain (Figure 4C,D), with particular emphasis on the lipid tail (Figure 4E,F), reinforcing the hypothesis of deep insertion into the membrane interface as suggested by liquid state NMR data. Other linear cationic amphipathic peptides such as magainin, HB43, or designed model sequences have also shown considerable disordering of the fatty acyl chain packing when intercalating into the membrane interface [91,93,94,95]. SAAP-148 shows a particularly pronounced effect with an up to >30% reduction in the order parameter (Figure 4E,F). Furthermore, in these previous cases, a preferential disordering of the PG palmitoyl chain has been observed [91,93].

### 3.3. MD Simulations of SAAP-148 Interacting with Biomimetic Bilayers

In order to better understand the details of the interaction of SAAP-148 with its targeted membranes, we complemented our NMR results with several molecular dynamics simulations. To this end, we have simulated membrane systems containing phospholipid ratios representative of bacterial and mammalian cells [79], to distinguish between their bactericidal and possibly toxic effects, respectively.

One of the simplest models of a mammalian cell membrane consists of POPC phospholipids. Despite its simplicity, this system has been used in a wide range of studies in which it was able to successfully reproduce biological effects [16,96,97,98]. This is due to the fact that the PC moiety is the main headgroup exposed on mammalian cells, while other groups such as PE are often exposed to the inner leaflet [79,99,100,101]. In the present work, we have also considered the effect of cholesterol, another key lipid in mammalian cell membranes.

Regarding bacteria, as mentioned above, their cell membranes usually expose anionic lipids such as PG and CL [78,79] but also PE. As a consequence, we have extensively focused on PE/PG or PE/PG/CL bilayers and also on their pure components PE, PG, or CL to isolate their contribution to the interaction in the mixtures.

Since most media used for the determination of minimal inhibitory concentrations contain Na^+^, Mg^2+^, and Ca^2+^, simulations were performed with each of these ions one at a time. Calcium is known to affect the interaction of peptides and proteins with the bilayers by modulating the interplay between self-aggregation and membrane binding [102,103]. Calcium can either promote aggregation [102] or binding [104,105,106,107]. In the first case, Ca^2+^ binds to the phosphate groups of the bilayer and shields its negative charge, thus preventing the binding of positively charged residues of the peptide. Probably for the same reason, Mg^2+^ is known to impair the action of some AMPs at high concentrations [108]. Less commonly, Ca^2+^ forms salt bridges connecting charged glutamate residues of the peptide to phosphate groups of the membrane, facilitating the interaction [102,107].

#### 3.3.1. Effect of SAAP-148 on Mammalian Model Membranes

In order to highlight the occurrence of key interatomic interactions triggering the association of SAAP-148 with membranes, we estimated the occurrence of H bonds, salt bridges, and van der Waals contacts along the trajectory (Figure 5A–C). This was quantified by calculating the radial distribution function [64] of all peptide atoms from each phospholipid atom. While polar contacts (H bonds and salt bridges) are particularly informative to study the first stages of the interaction, van der Waals contacts mostly inform on the final stages, in case the peptide has been inserted into the membrane.

While SAAP-148 is able to establish several polar contacts with POPC membranes (Appendix A), its penetration in the membrane remains superficial as testified by the infrequent apolar contacts (Appendix A). When the model is improved with the addition of cholesterol in the ratio POPC:CHO 70:30, the number of contacts is drastically reduced (Appendix A), and the preservation of the order parameter (Appendix A) indicates that the membrane interior is not affected by the presence of the peptide (with one exception in the presence of Mg^2+^). Interestingly, in the presence of cholesterol, we observe a loss of the alpha-helix fold at both peptide termini, which might be due to the selectivity of SAAP-148 towards bacterial membranes (Appendix A).

#### 3.3.2. Effect of SAAP-148 on Bacterial Model Membranes

With membrane models containing lipids characteristic of bacteria (PE, PG, CL), the occurrence of polar contacts increases by one order of magnitude as compared to mammalian models (compare Appendix A with Appendix A). Arginines and lysines drive the interaction by forming salt bridges with phosphate oxygen atoms of phospholipids (Figure 5B) following a clear preference order CL > PG > PE (Appendix A), which is probably based on the increasing negative charge of the target (−2, −1, 0).

It should be noted that the insertion of SAAP-148 into bacterial models requires long simulation times (up to 6 μs) in order to observe penetration of the peptide, which is observed only in rare cases, mostly in the presence of calcium. Following the pattern observed for other AMPs [14,15,87,109,110], the peptide first sticks to the membrane from its exterior by the formation of the salt bridges described above and rigidifying the motion of acyl chains as observed by higher values of their order parameter (Appendix A). Subsequently, the peptide fluidifies the membrane (Appendix A) by deforming the surface but not necessarily being fully internalized, thus reproducing in some repetitions the experimental behavior of the order parameter (compare Figure 4C,D with Appendix A). Although a significant reduction in the order parameter along the acyl chain of phospholipids can be obtained even without complete peptide internalization, we believe that SAAP-148 does get internalized in longer time scales (which might require enhanced sampling methods [111]). The orientation of the helix obtained by paramagnetic effects (where apolar side chains are closer to the center of micelles than charged ones) and the large effect on the order parameter of the very end of lipid tails (Figure 4E,F) suggest a deep internalization.

The significant perturbation of the surface can be monitored by changes in the electron density profile (Appendix A) and dipole potential profiles (Appendix A). The former (see Appendix A) shows a clear inter-digitalization of the acyl chain extremities (increased density in the central part of the membrane) and a splitting of the phosphate maxima (at −2 and +2 nm from the center) in two populations due to disorder induced by the presence of the peptide and the entrance of water (whose density is not included in the graph). The latter (see Appendix A) indicates that the presence of SAAP-148 lowers the dipole potential, thus facilitating the destabilizing transit of dipolar molecules across the bilayer [79,112,113].

Although a complete penetration of the peptide was observed only in simulations with a single peptide (Figure 5D), visual examination of simulations with multiple peptides suggests that dimerization could be an important phenomenon for the interaction (Figure 5E, Appendix A), although the dimer poorly penetrates the bilayers, at least in the time scale of our simulations (Appendix A). Dimerization of SAAP-148 would not be surprising for a peptide that in helical conformation forms a perfect amphipathic helix and is perfectly consistent with what is observed experimentally in NMR spectra of the free peptide (see Section 3.1.1). Simulations show that, as long as the two hydrophobic faces shield each other from exposure to water, parallel or antiparallel alignment are both possible (Figure 5E).

## 4. Conclusions

SAAP-148 is an extremely promising antimicrobial peptide due to its capacity to kill MDR bacteria, prevent the formation of their biofilm, and even eradicate persister cells [31]. Furthermore, it has been shown to be resistant to degradation in physiological fluids [31]. The antimicrobial activity of SAAP-148 has been explained by its ability to interact and destabilize the bacterial membrane. This is reflected by the peptide’s ability to disrupt liposomes containing bacterial phospholipids [31,33] and potentiate the activity of other antibiotics [12]. In this work, we have elucidated its mechanism of action at the molecular level of detail. Experiments based on liquid and solid-state NMR spectroscopies show that SAAP-148 forms stable alpha-helical structures on the membrane surface and is able to become internalized in bacterial membrane mimetics. Molecular dynamics simulations require a long time to observe this internalization, indicating that the energy barrier is higher than for other AMPs [94]. Such a barrier could be lowered by the presence of metal ions such as Ca^2+^, although further studies are needed to verify this phenomenon.

Most importantly, using a variety of complementary approaches, we have consistently shown that the axis of the helix formed by SAAP-148 on bacterial-like membranes is almost perpendicular to the bilayer normal. Thus, SAAP-148 can act by breaking the membrane once a dense peptide carpet has been accumulated at the membrane interface, rather than forming transmembrane pores. This is consistent with the SMART model [34], in which amphipathic helices lay on the surface. At low peptide:lipid ratios, the soft bilayers can adjust to the disturbance caused by the interfacial peptide. According to the model, the bilayer packing can even be stabilized due the presence of peptide, e.g., in PE-rich membranes [114] (see the increase in the order parameter in Appendix A). At higher peptide:lipid ratios, the changes in the order parameter are consistent with a disruption in the membrane order (Figure 4C–F and one repetition of simulations in Appendix A) leading to the formation of transient pores [34]. This allows the internalization of the peptide and the subsequent formation of membrane-disassembling structures.

## Figures and Tables

**Figure 1 pharmaceutics-15-00761-f001:**
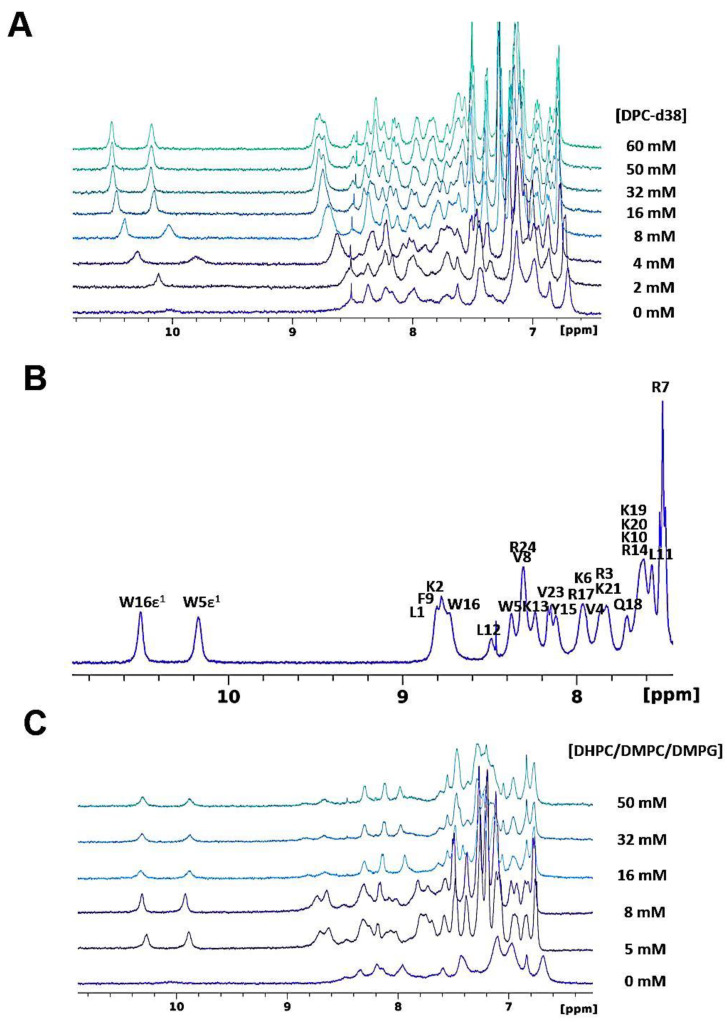
(**A**) The 500 MHz ^1^H NMR spectra (amide and aromatic region) recording during the titration of SAAP-148 with DPC:d38 micelles up to a concentration of 60 mM. (**B**) NMR assignment of the amide region of SAAP-148 in DPC micelles. (**C**) The 500 MHz ^1^H NMR spectra of the titration of SAAP-148 with DHPC/DMPC/DMPG isotropic bicelles up to a concentration of 50 mM. The temperature was 310 K.

**Figure 2 pharmaceutics-15-00761-f002:**
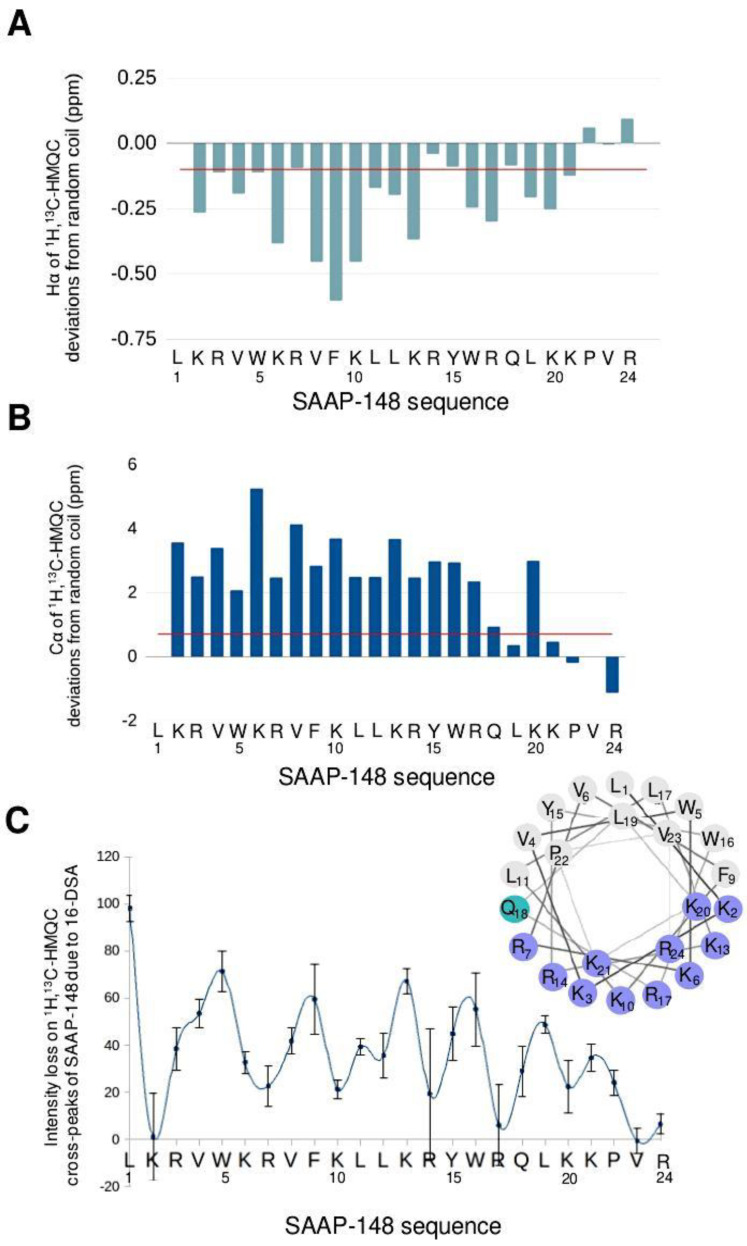
(**A**,**B**) Hα (**A**) and Cα (**B**) chemical shift deviations from random coil values (see Materials and Methods). (**C**) Percental intensity loss of Hα/Cα ^1^H,^13^C-HMQC peaks in the presence of paramagnetic 16-DSA. Insert shows a helical wheel for SAAP-148 peptide (created using Netwheels web server [88]).

**Figure 3 pharmaceutics-15-00761-f003:**
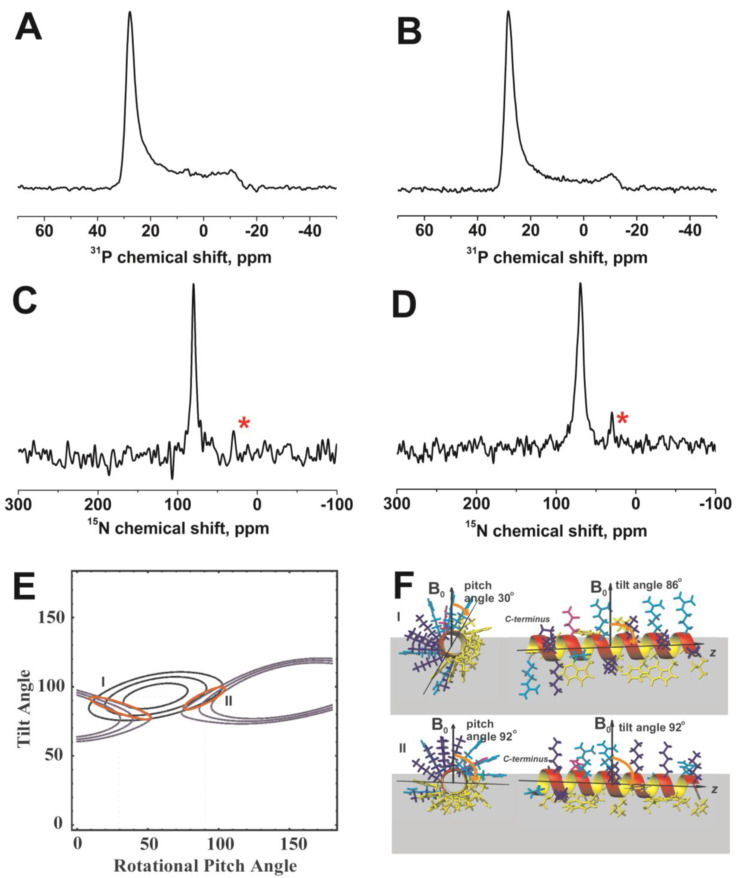
Proton-decoupled ^31^P (**A**,**B**) and ^15^N (**C**,**D**) spectra of 2 mol% SAAP-148 carrying ^15^N label in position Leu11 (**A**,**C**) and Leu12 (**B**,**D**) in POPE:POPG 3:1 at 310 K. The ^15^N signal from the PE headgroup at natural isotope abundance is indicated by asterisks in panels (**C**) and (**D**). (**E**) Restriction analysis and corresponding peptide alignments relative to the membrane normal (**F**) for 2 mol% SAAP-148 in POPE:POPG 3:1 bilayer. The resulting restrictions for SAAP-148 topology are shown in panel E: ^15^N-Leu11 is 80.0 ± 2.5 ppm in blue, and ^15^N-Leu12 is 70.0 ± 4.0 ppm in green. The restraints were obtained assuming rocking (18° Gaussian distribution) and wobbling (10°) motions of an ideal helix (φ = −65°, ψ = −45°). Leucines, phenylalanines, valines, tryptophan, and proline are shown in yellow, lysines in blue, glutamines in magenta, and arginines in cyan. Two possible intersections are obtained in (**E**) (orange ellipses), both are represented in panel (**F**) (note that the perfect helix shown in panel F does not represent an experimental structure but a way to show the axis orientation in the assumption of alpha helix conformation).

**Figure 4 pharmaceutics-15-00761-f004:**
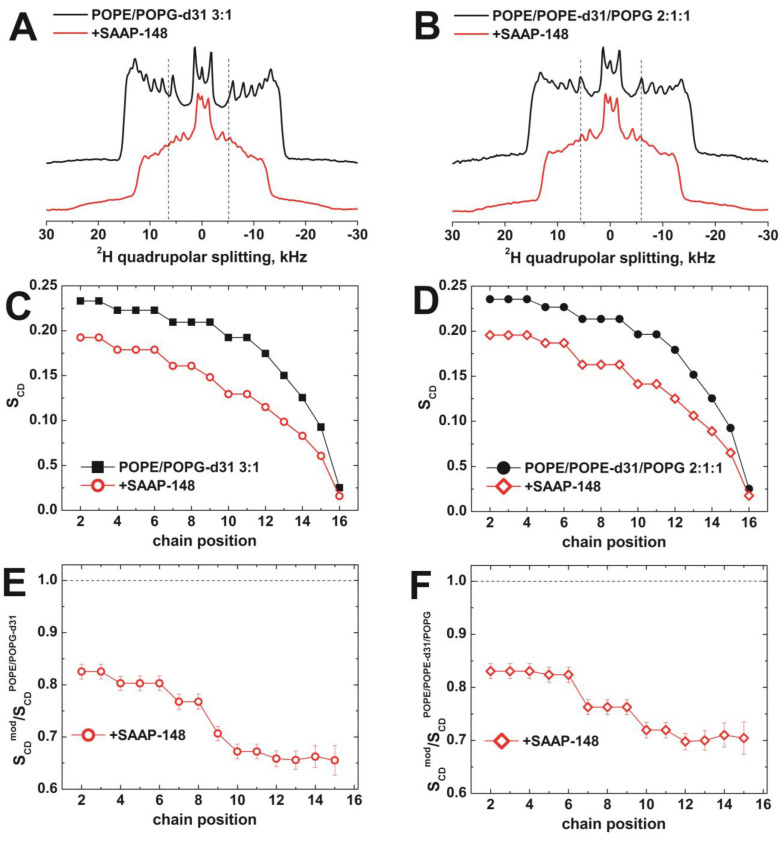
(**A**,**B**) Quadrupolar ^2^H solid-state NMR spectra of chain-deuterated POPG-d_31_ (**A**) or POPE-d_31_ (**B**) in the presence or absence of SAAP-148 peptide at 2 mol% concentration at 310 K. Samples were hydrated with Tris buffer pH 7.4 to reach a water content of h = 0.81. (**C**,**D**) Order parameters S_CD_ of lipid palmitoyl chain C–D bonds in the absence and in the presence of SAAP-148 for POPG-d_31_ (**C**) and POPE-d_31_ (**D**). The effect of SAAP-148 is more clearly evidenced when plotting the ratio of S_CD_ in the presence of SAAP-148 with that in its absence (**E**,**F**).

**Figure 5 pharmaceutics-15-00761-f005:**
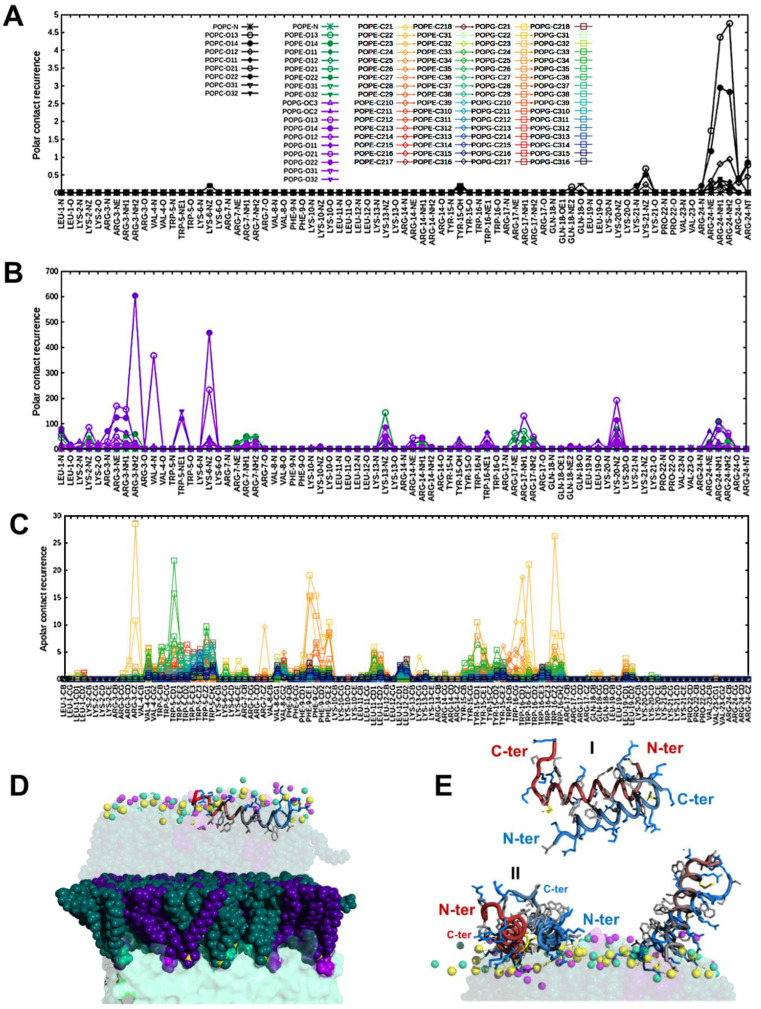
(**A**,**B**) Recurrence of polar atom contacts (H bonds and salt bridges) between SAAP-148 peptide and POPC (**A**) and POPE/POPG (**B**) bilayers calculated along MD simulation trajectories. (**C**) Recurrence of van der Waals contacts between SAAP-148 peptide and POPE/POPG bilayers calculated along MD simulation trajectories. (**D**) MD snapshot of SAAP-148 interacting with POPE/POPG bilayers in the presence of Ca^2+^ ion. Color code: phosphorus atom: yellow, POPE: dark green (body), turquoise (headgroup), light green (amine of the headgroup); POPG: dark violet (body), violet (headgroup), light violet (hydroxyls of the headgroup). For clarity, only functional groups of headgroups are shown (spheres) in the upper leaflet. SAAP-148 is shown as a ‘tube’ colored from blue (N-terminus) to red (C-terminus). Sidechains are shown as sticks with the following color code: positively charged (blue) and nonpolar (light gray). (**E**) SAAP-148 dimers found along the trajectory and interacting with POPE/POPG bilayers. In the dimer, the two units (colored in red and blue) can arrange in antiparallel (I) or parallel fashion (II).

## Data Availability

Not applicable.

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
