# Peer review of "The Mechanism of Action of SAAP-148 Antimicrobial Peptide as Studied with NMR and Molecular Dynamics Simulations"

_pharmaceutics, 2023, doi:10.3390/pharmaceutics15030761_

Round 1
Reviewer 1 Report
The manuscript entitled “The mechanism of action of SAAP-148 antimicrobial peptide as studied by NMR and molecular dynamics simulations” evaluate the mechanism of action of the peptide SAAP-148 (peptide derived from LL-37) by liquid and solid-state NMR spectroscopies in micelle, bicelle and bacterial biomimetic membranes. In my opinion, the manuscript is interesting, but it needs to be improved before publication.
A revision of the English language should be carried out.
Ex:
In page 2: The sentence “Samples were typically at a concentration of 2.4 mM.” is incomplete
In page 7: The sentence “The main limitation of DPC resides in the fact that micelles” is incomplete.
In page 2: “The carboxyl amide c-terminus”, fix the letter “c” (“C”).
In page 3: “2 mole%”
In page 8: The sentence “In the case of SAAP-148 we placed a paramagnetic center in the center of DPC micelles” needs to be clear.
In abstract is putted that “thus probably forming a carpet at the bacterial membrane rather than a pore.”. The carpet is an action mechanism? The carpet is the first step for mechanism of action called “carpet like”.
The characterization of the peptide (HPLC profiles and mass spectra) needs to be added in Supplementary material.
In introduction is putted that “The maximum killing rate of AMPs is usually much higher than that of antibiotics, making it harder for the resistance mechanisms to be developed”. The difficulty of the resistance mechanism is also due to the mechanism of action. Comment?
The better description of the peptide SAAP-148 is necessary.
In introduction in putted that the carpet-model causes the formation of transient micro-sized pores. Is it right?
At the end of page 7 it says "alpha helix interrupted only by proline", but couldn't the alpha helix be less stable in the C-terminal region due to the decrease in hydrogen bonds?
The Figure 3A is cited after Figure 3B. Fixed.
In page 10, is putted that “shown in Figure 3E as blue (Leu 11) and green lines (Leu 12)”. I can't see green lines in Figure 3E.
In Figure 3F, I can´t see where the alpha helix interrupted only by proline. Explain.
In page 7 is putted that “In helical conformation, this peptide would form a perfectly amphipathic helix, suggesting that it might dimerize to avoid exposure of apolar side chain to the water”, but CD spectra do not indicate alpha-helix structure. Explain.
In figure 5: Where is POPC black (body)?
In conclusion, the phrase “Such a barrier could be lowered by the presence of metal ions like Ca2+, although further studies are needed to verify this phenomenon.” is not supported by results.
In page 14 is putted that the dimerization occurs between two peptides. Is possible that more peptide chains interact to form one barrel stave pore?
Reviewer 2 Report
Adélaïde et al., presented antimicrobial peptide is nice piece of work. I am satisfied with the results and presentation of results in the manuscript. I found some unevenness in the reference section such as font, bold, journal name. I recommend author to correct that in even format. Overall, I do not find any flaws in the manuscript.
Reviewer 3 Report
The authors investigated the antimicrobial peptide SAAP-148 to understand its mode of interaction using NMR techniques, including DPC micelles in solution NMR and liposomes in solid state NMR. Additionally, molecular dynamics simulations were performed to gain a deeper understanding of lipid-peptide interactions. The main conclusion of the study is that the peptide rarely inserts into the membrane and instead acts superficially.
The manuscript is interesting and important, with only minor comments.
For the reader's benefit, it would be helpful to provide some basic information about the peptide's characteristics, such as its minimum inhibitory concentration (MIC) values against model strains and toxicity indicators.
On page 1, the authors write: "The maximum killing rate of AMPs is usually much higher than that of antibiotics, making it harder for resistance mechanisms to be developed." The killing rate does not suppress the development of resistance, as this is achieved through group selection, not the evolution of a single organism.
Lastly, the abbreviation DPC micelle is not explained.
Round 2
Reviewer 1 Report
The manuscript entitled “The mechanism of action of SAAP-148 antimicrobial peptide as studied by NMR and molecular dynamics simulations”, in my opinion, is interesting and could be acceptable by publication.
In the figure S1, the analytical conditions could be added.
Author Response
We thank the referee for the appreciation of our work and the advice to add more information in the caption of Figure S1.
The caption now specifies the type of column used por the HPLC purification and the composition of the mobile phase.